# M-Best-Diverse Labelings
# for Submodular Energies and Beyond

**Alexander Kirillov**[1]      **Dmitrij Schlesinger**[1]      **Dmitry Vetrov**[2]
**Carsten Rother**[1]      **Bogdan Savchynskyy**[1]
[1] TU Dresden, Dresden, Germany      [2] Skoltech, Moscow, Russia
`alexander.kirillov@tu-dresden.de`

## Abstract

We consider the problem of finding $M$ best diverse solutions of energy minimization problems for graphical models. Contrary to the sequential method of Batra et al., which greedily finds one solution after another, we infer all $M$ solutions jointly. It was shown recently that such jointly inferred labelings not only have smaller total energy but also qualitatively outperform the sequentially obtained ones. The only obstacle for using this new technique is the complexity of the corresponding inference problem, since it is considerably slower algorithm than the method of Batra et al. In this work we show that the joint inference of $M$ best diverse solutions can be formulated as a submodular energy minimization if the original MAP-inference problem is submodular, hence fast inference techniques can be used. In addition to the theoretical results we provide practical algorithms that outperform the current state-of-the-art and can be used in both submodular and non-submodular case.

## 1  Introduction

A variety of tasks in machine learning can be formulated in the form of an energy minimization problem, known also as *maximum a posteriori* (MAP) or *maximum likelihood estimation* (MLE) inference in an undirected graphical models (related to Markov or conditional random fields). Its modeling power and importance are well-recognized, which resulted into specialized benchmark, i.e. [18] and computational challenges [8] for its solvers. This underlines the importance of finding *the most probable* solution. Following [3] and [25] we argue, however, that finding $M > 1$ *diverse* configurations with low energies is also of importance in a number of scenarios, such as: (a) Expressing uncertainty of the found solution [27]; (b) Faster training of model parameters [14]; (c) Ranking of inference results [32]; (d) Empirical risk minimization [26].

We build on *the new formulation* for finding $M$-best-diverse-configurations, which was recently proposed in [19]. In this formulation all $M$ configurations are inferred *jointly*, contrary to the established method [3], where a sequential greedy procedure is used. As shown in [19], the new formulation does not only reliably produce configurations with lower total energy, but also leads to better results in several application scenarios. In particular, for the image segmentation scenario the results of [19] significantly outperform those of [3]. This is true even when [19] uses a plain Hamming distance as a diversity measure and [3] uses more powerful diversity measures.

**Our contributions.**
    • We show that finding $M$-best-diverse configurations of a binary submodular energy minimization can be formulated as a submodular MAP-inference problem, and hence can be solved

______________________

    This project has received funding from the European Research Council (ERC) under the European Unions Horizon 2020 research and innovation programme (grant agreement No 647769). D. Vetrov was supported by RFBR proj. (No. 15-31-20596) and by Microsoft (RPD 1053945).

efficiently for *any* node-wise diversity measure.

 • We show that for certain diversity measures, such as e.g. Hamming distance, the $M$-best-diverse configurations of a multilabel submodular energy minimization can be formulated as a submodular MAP-inference problem, which also implies applicability of efficient graph cut-based solvers.

 • We give the insight that if the MAP-inference problem is submodular then the $M$-best-diverse configurations can be always fully ordered with respect to the natural partial order, induced in the space of all configurations.

 • We show experimentally that if the MAP-inference problem is submodular, we are quantitatively at least as good as [19] and considerably better than [3]. The main advantage of our method is a major speed up over [19], up to the order of two magnitudes. Our method has the same order of magnitude run-time as [3]. In the non-submodular case our results are slightly inferior to [19], but the advantage with respect to gain in speed up still holds.

**Related work.** The importance of the considered problem may be justified by the fact that a procedure of computing $M$-*best solutions* to discrete optimization problems was proposed in [23], which dates back to 1972. Later, more efficient specialized procedures were introduced for MAP-inference on a tree [29, Ch. 8], junction-trees [24] and general graphical models [33, 12, 2]. Such methods are however not suited for scenarios where diversity of the solutions is required (like in machine translation, search engines, producing $M$-best hypothesis in cascaded algorithms), since they do not enforce it explicitly.

*Structural Determinant Point Processes* [22] is a tool to model probabilistic distributions over structured models. Unfortunately an efficient sampling procedure is feasible for tree-structured graphical models only. The recently proposed algorithm [7] to find $M$ *best modes* of a distribution is limited to the same narrow class of problems.

Training of $M$ *independent* graphical models to produce diverse solutions was proposed in [13, 15]. In contrast, we assume *a single fixed* model supporting reasonable MAP-solutions.

Along with [3], the most related to our work is the recent paper [25], which proposes a subclass of new diversity penalties, for which the greedy nature of the algorithm [3] can be substantiated due to submodularity of the used diversity measures. In contrast to [25] we do not limit ourselves to diversity measures fulfilling such properties and moreover, we define a class of problems, for which our joint inference approach leads to polynomially and *efficiently* solvable problems in practice.

We build on top of the work [19], which is explained in detail in Section 2.

**Organization of the paper.** Section 2 provides background necessary for formulation of our results: energy minimization for graphical models and existing approaches to obtain diverse solutions. In Section 3 we introduce submodularity for graphical models and formulate the main results of our work. Finally, Section 4 and 5 are devoted to the experimental evaluation of our technique and conclusions. Supplementary material contains proofs of all mathematical claims and the concurrent submission [19].

## 2   Preliminaries

**Energy minimization.** Let $2^{\mathcal{A}}$ denote the powerset of a set $\mathcal{A}$. The pair $\mathcal{G} = (\mathcal{V}, \mathcal{F})$ is called a *hyper-graph* and has $\mathcal{V}$ as a finite *set of variable nodes* and $\mathcal{F} \subseteq 2^{\mathcal{V}}$ as *a set of factors*. Each variable node $v \in \mathcal{V}$ is associated with a *variable* $y_v$ taking its values in a finite *set of labels* $L_v$. The set $L_{\mathcal{A}} = \prod_{v \in \mathcal{A}} L_v$ denotes a Cartesian product of sets of labels corresponding to the subset $\mathcal{A} \subseteq \mathcal{V}$ of variables. Functions $\theta_f \colon L_f \to \mathbb{R}$, associated with factors $f \in \mathcal{F}$, are called *potentials* and define local costs on values of variables and their combinations. Potentials $\theta_f$ with $|f| = 1$ are called *unary*, with $|f| = 2$ *pairwise* and $|f| > 2$ *higher order*. The set $\{\theta_f \colon f \in \mathcal{F}\}$ of all potentials is referred by $\boldsymbol{\theta}$. For any factor $f \in \mathcal{F}$ the corresponding set of variables $\{y_v \colon v \in f\}$ will be denoted by $y_f$. *The energy minimization* problem consists of finding *a labeling* $\boldsymbol{y}^* = \{y_v \colon v \in \mathcal{V}\} \in L_{\mathcal{V}}$, which minimizes the total sum of corresponding potentials:

$$\boldsymbol{y}^* = \arg \min_{\boldsymbol{y} \in L_{\mathcal{V}}} E(\boldsymbol{y}) = \arg \min_{\boldsymbol{y} \in L_{\mathcal{V}}} \sum_{f \in \mathcal{F}} \theta_f(y_f) \,. \tag{1}$$

Problem (1) is also known as *MAP-inference*. Labeling $\boldsymbol{y}^*$ satisfying (1) will be later called *a solution of the energy-minimization* or *MAP-inference problem*, shortly *MAP-labeling* or *MAP-solution*.

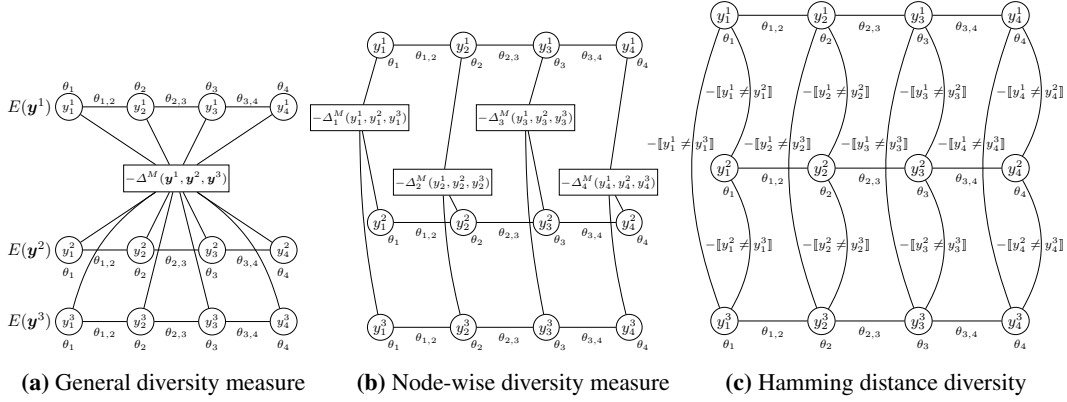

**(a)** General diversity measure     **(b)** Node-wise diversity measure     **(c)** Hamming distance diversity

Figure 1: Examples of factor graphs for 3 diverse solutions of the original MRF (1) with different diversity measures. The circles represent nodes of the original model that are copied 3 times. For clarity the diversity factors of order higher than 2 are shown as squares. Pairwise factors are depicted by edges connecting the nodes. We omit $\lambda$ for readability. (a) The most general diversity measure (4), (b) the node-wise diversity measure (6), (c) Hamming distance as a diversity measure (5).

Finally, a *model* is defined by the triple $(\mathcal{G}, \mathcal{L}_{\mathcal{V}}, \boldsymbol{\theta})$, i.e. the underlying hyper-graph, the sets of labels and the potentials.

In the following, we use brackets to distinguish between upper index and power, i.e. $(\mathcal{A})^n$ means the $n$-th power of $\mathcal{A}$, whereas $n$ is an upper index in the expression $\mathcal{A}^n$. We will keep, however, the standard notation $\mathbb{R}^n$ for the $n$-dimensional vector space.

**Sequential Computation of $M$ Best Diverse Solutions [3].** Instead of looking for a single labeling with lowest energy, one might ask for a set of labelings with low energies, yet being significantly different from each other. In order to find such $M$ diverse labelings $\boldsymbol{y}^1, \ldots, \boldsymbol{y}^M$, the method proposed in [3] solves a sequence of problems of the form

$$\boldsymbol{y}^m = \arg\min_{\boldsymbol{y}} \left[ E(\boldsymbol{y}) - \lambda \sum_{i=1}^{m-1} \Delta(\boldsymbol{y}, \boldsymbol{y}^i) \right] \tag{2}$$

for $m = 1, 2 \ldots, M$, where $\lambda > 0$ determines a trade-off between diversity and energy, $\boldsymbol{y}^1$ is the MAP-solution and the function $\Delta \colon L_{\mathcal{V}} \times L_{\mathcal{V}} \to \mathbb{R}$ defines the *diversity* of two labelings. In other words, $\Delta(\boldsymbol{y}, \boldsymbol{y}')$ takes a large value if $\boldsymbol{y}$ and $\boldsymbol{y}'$ are diverse, in a certain sense, and a small value otherwise. This problem can be seen as an energy minimization problem, where additionally to the initial potentials $\boldsymbol{\theta}$ the potentials $-\lambda\Delta(\cdot, \boldsymbol{y}^i)$, associated with an additional factor $\mathcal{V}$, are used. In the simplest and most commonly used form, $\Delta(\boldsymbol{y}, \boldsymbol{y}')$ is represented by a sum of node-wise diversity measures $\Delta_v \colon L_v \times L_v \to \mathbb{R}$,

$$\Delta(\boldsymbol{y}, \boldsymbol{y}') = \sum_{v \in \mathcal{V}} \Delta_v(y_v, y'_v) \,, \tag{3}$$

and the potentials are split to a sum of *unary* potentials, i.e. those associated with additional factors $\{v\}$, $v \in \mathcal{V}$. This implies that in case efficient graph-cut based inference methods (including $\alpha$-expansion [6], $\alpha$-$\beta$-swap [6] or their generalizations [1, 10]) are applicable to the initial problem (1) then they remain applicable to the augmented problem (2), which assures efficiency of the method.

**Joint computation of M-best-diverse labelings.** The notation $f^M(\{\boldsymbol{y}\})$ will be used as a shortcut for $f^M(\boldsymbol{y}^1, \ldots, \boldsymbol{y}^M)$, for any function $f^M \colon (L_{\mathcal{V}})^M \to \mathbb{R}$.

Instead of the greedy sequential procedure (2), in [19] it was suggested to infer all $M$ labelings *jointly*, by minimizing

$$E^M(\{\boldsymbol{y}\}) = \sum_{i=1}^{M} E(\boldsymbol{y}^i) - \lambda\Delta^M(\{\boldsymbol{y}\}) \tag{4}$$

for $\boldsymbol{y}^1, \ldots, \boldsymbol{y}^M$ and some $\lambda > 0$. Function $\Delta^M$ defines *the total* diversity of any $M$ labelings.

It was shown in [19] that the $M$ labelings obtained according to (4) have both lower total energy $\sum_{i=1}^{M} E(\boldsymbol{y}^i)$ and are better from the applied point of view, than those obtained by the sequential method (2). Hence we will build on the formulation (4) in this work.

Though the expression (4) looks complicated, it can be nicely represented in the form (1) and hence constitutes an energy minimization problem. To achieve this, one creates $M$ copies $(\mathcal{G}^i, \mathcal{L}_\mathcal{V}^i, \boldsymbol{\theta}^i) = (\mathcal{G}, \mathcal{L}_\mathcal{V}, \boldsymbol{\theta})$ of the initial model $(\mathcal{G}, \mathcal{L}_\mathcal{V}, \boldsymbol{\theta})$. The hyper-graph $\mathcal{G}_1^M = (\mathcal{V}_1^M, \mathcal{F}_1^M)$ for the new task is defined as follows. The set of nodes in the new graph is the union of the node sets from the considered copies $\mathcal{V}_1^M = \bigcup_{i=1}^M \mathcal{V}^i$. Factors are $\mathcal{F}_1^M = \bigcup_{i=1}^M \mathcal{F}^i \cup \{\mathcal{V}_1^M\}$, i.e. again the union of the initial ones extended by a special factor corresponding to the diversity penalty that depends on all nodes of the new graph. Each node $v \in \mathcal{V}^i$ is associated with the label set $L_v^i = L_v$. The corresponding potentials $\boldsymbol{\theta}_1^M$ are defined as $\{-\lambda \Delta^M, \boldsymbol{\theta}^1, \dots, \boldsymbol{\theta}^M\}$, see Fig. 1a for illustration. The model $(\mathcal{G}_1^M, \mathcal{L}_{\mathcal{V}_1^M}, \boldsymbol{\theta}_1^M)$ corresponds to the energy (4). An optimal $M$-tuple of these labelings, corresponding to a minimum of (4), is a trade-off between low energy of individual labelings $\boldsymbol{y}^i$ and their total diversity.

**Complexity of the Diversity Problem** (4). Though the formulation (4) leads to better results than those of (2), minimization of $E^M$ is computationally demanding even if the original energy $E$ can be easily (approximatively) optimized. This is due to the intrinsic *repulsive* structure of the diversity potentials $-\lambda \Delta^M$: according to the intuitive meaning of the diversity, similar labels are penalized more than different one. Consider the simplest case with the Hamming distance applied node-wise as a diversity measure

$$\Delta^M(\{\boldsymbol{y}\}) = \sum_{i=1}^{M-1} \sum_{j=i+1}^{M} \sum_{v \in \mathcal{V}} \Delta_v(y_v^i, y_v^j), \text{ where } \Delta_v(y, y') = [\![y \neq y']\!].\tag{5}$$

Here expression $[\![A]\!]$ equals 1 if $A$ is true and 0 otherwise. The corresponding factor graph is sketched in Fig. 1c. Such potentials can not be optimized with efficient graph-cut based methods and moreover, as shown in [19], the bounds delivered by LP-relaxation [31] based solvers are very loose in practice. Indeed, solutions delivered by such solvers are significantly inferior even to the results of the sequential method (2).

To cope with this issue *a clique encoding representation* of (4) was proposed in [19]. In this representation $M$-tuples of labels $y_v^1, \dots, y_v^M$ (in the $M$ nodes corresponding to the single initial node $v$) were considered as the new labels. In this way the difficult diversity factors were incorporated into the unary factors of the new representation and the pairwise factors were adjusted respectively. This allowed to (approximately) solve the problem (4) with graph-cuts based techniques if those techniques were applicable to the energy $E$ of a single labeling. The disadvantage of the clique encoding representation is the exponential growth of the label space, which was reflected in a significantly higher inference time for the problem (4) compared to the procedure (2). In what follows, we show an alternative transformation of the problem (4), which (i) does not have this drawback (its size is basically the same as those of (4)) and (ii) allows to *exactly* solve (4) in the case the energy $E$ is submodular.

**Node-wise Diversity.** In what follows we will mainly consider *the node-wise diversity measures*, i.e. those, which can be represented in the form

$$\Delta^M(\{\boldsymbol{y}\}) = \sum_{v \in \mathcal{V}} \Delta_v^M(\{\boldsymbol{y}\}_v)\tag{6}$$

for some *node diversity measures* $\Delta_v^M: (L_v)^M \to \mathbb{R}$, see Fig. 1b for illustration.

## 3  M-Best-Diverse Labelings for Submodular Problems

**Submodularity.** In what follows we will assume that the sets $L_v$, $v \in \mathcal{V}$, of labels are completely ordered. This implies that for any $s, t \in \mathcal{L}_v$ their maximum and minimum, denoted as $s \vee t$ and $s \wedge t$ respectively, are well-defined. Similarly let $\boldsymbol{y}_1 \vee \boldsymbol{y}_2$ and $\boldsymbol{y}_1 \wedge \boldsymbol{y}_2$ denote the node-wise maximum and minimum of any two labelings $\boldsymbol{y}_1, \boldsymbol{y}_2 \in L_\mathcal{A}$, $\mathcal{A} \subseteq \mathcal{V}$. Potential $\theta_f$ is called *submodular*, if for any two labelings $\boldsymbol{y}_1, \boldsymbol{y}_2 \in L_f$ it holds[1]:

$$\theta_f(\boldsymbol{y}_1) + \theta_f(\boldsymbol{y}_2) \geq \theta_f(\boldsymbol{y}_1 \vee \boldsymbol{y}_2) + \theta_f(\boldsymbol{y}_1 \wedge \boldsymbol{y}_2).\tag{7}$$

Potential $\theta$ will be called *supermodular*, if $(-\theta)$ is submodular.

Energy $E$ is called submodular if for any two labelings $\boldsymbol{y}_1, \boldsymbol{y}_2 \in L_{\mathcal{V}}$ it holds:

$$E(\boldsymbol{y}_1) + E(\boldsymbol{y}_2) \geq E(\boldsymbol{y}_1 \vee \boldsymbol{y}_2) + E(\boldsymbol{y}_1 \wedge \boldsymbol{y}_2). \tag{8}$$

Submodularity of energy trivially follows from the submodularity of all its non-unary potentials $\theta_f$, $f \in \mathcal{F}, |f| > 1$. In the pairwise case the inverse also holds: submodularity of energy implies also submodularity of all its (pairwise) potentials (e.g. [31, Thm. 12]). There are efficient methods for solving energy minimization problems with submodular potentials, based on its transformation into min-cut/max-flow problem [21, 28, 16] in case all potentials are either unary or pairwise or to a submodular max-flow problem in the higher-order case [20, 10, 1].

**Ordered $M$ Solutions.** In what follows we will write $\boldsymbol{z^1} \leq \boldsymbol{z^2}$ for any two vectors $\boldsymbol{z}^1$ and $\boldsymbol{z^2}$ meaning that the inequality holds coordinate-wise.

For an arbitrary set $\mathcal{A}$ we will call a function $f \colon (\mathcal{A})^n \rightarrow \mathbb{R}$ of $n$ variables *permutation invariant* if for any $(x^1, x^2, \dots, x^n) \in (\mathcal{A})^n$ and any permutation $\pi$ it holds $f(x^1, x^2, \dots, x^n) = f(x^{\pi(1)}, x^{\pi(2)}, \dots, x^{\pi(n)})$. In what follows we will consider mainly permutation invariant diversity measures.

Let us consider two arbitrary labelings $\boldsymbol{y}^1, \boldsymbol{y}^2 \in L_{\mathcal{V}}$ and their node-wise minimum $\boldsymbol{y}^1 \wedge \boldsymbol{y}^2$ and maximum $\boldsymbol{y}^1 \vee \boldsymbol{y}^2$. Since $(y_v^1 \wedge y_v^2, y_v^1 \vee y_v^2)$ is either equal to $(y_v^1, y_v^2)$ or to $(y_v^2, y_v^1)$, for any permutation invariant node diversity measure it holds $\Delta_v^2(y_v^1, y_v^2) = \Delta_v^2(y_v^1 \wedge y_v^2, y_v^1 \vee y_v^2)$. This in its turn implies $\Delta^2(\boldsymbol{y}^1 \wedge \boldsymbol{y}^2, \boldsymbol{y}^1 \vee \boldsymbol{y}^2) = \Delta^2(\boldsymbol{y}^1, \boldsymbol{y}^2)$ for any node-wise diversity measure of the form (6). If $E$ is submodular, then from (8) it additionally follows that

$$E^2(\boldsymbol{y}^1 \wedge \boldsymbol{y}^2, \boldsymbol{y}^1 \vee \boldsymbol{y}^2) \leq E^2(\boldsymbol{y}^1, \boldsymbol{y}^2), \tag{9}$$

where $E^2$ is defined as in (4). Note, that $(\boldsymbol{y}^1 \wedge \boldsymbol{y}^2) \leq (\boldsymbol{y}^1 \vee \boldsymbol{y}^2)$. Generalizing these considerations to $M$ labelings one obtains

**Theorem 1.** *Let $E$ be submodular and $\Delta^M$ be a node-wise diversity measure with each component $\Delta_v^M$ being permutation invariant. Then there exists an ordered $M$-tuple $(\boldsymbol{y}^1, \dots, \boldsymbol{y}^M)$, $\boldsymbol{y}^i \leq \boldsymbol{y}^j$ for $1 \leq i < j \leq M$, such that for any $(\boldsymbol{z}^1, \dots, \boldsymbol{z}^M) \in (L_{\mathcal{V}})^M$ it holds*

$$E^M(\{\boldsymbol{y}\}) \leq E^M(\{\boldsymbol{z}\}), \tag{10}$$

*where $E^M$ is defined as in* (4).

Theorem 1 in particular claims that in the binary case $L_v = \{0, 1\}$, $v \in \mathcal{V}$, the optimal $M$ labelings define nested subsets of nodes, corresponding to the label 1.

**Submodular formulation of M-Best-Diverse problem.** Due to Theorem 1, for submodular energies and node-wise diversity measures it is sufficient to consider only ordered $M$-tuples of labelings.

This order can be enforced by modifying the diversity measure accordingly:

$$\hat{\Delta}_v^M(y^1, \dots, y^M) := \begin{cases} \Delta_v^M(y^1, \dots, y^M), & y^1 \leq y^2 \leq \dots \leq y^M \\ -\infty, & \text{otherwise} \end{cases}, \tag{11}$$

and using it instead of the initial measure $\Delta_v^M$. Note that $\hat{\Delta}_v^M$ is *not* permutation invariant. In practice one can use sufficiently big numbers in place of $\infty$ in (11). This implies

**Lemma 1.** *Let $E$ be submodular and $\Delta^M$ be a node-wise diversity measure with each component $\Delta_v^M$ being permutation invariant. Then any solution of* the ordering enforcing $M$-best-diverse prob-*lem*

$$\hat{E}^M(\{\boldsymbol{y}\}) = \sum_{i=1}^{M} E(\boldsymbol{y}^i) - \lambda \sum_{v \in \mathcal{V}} \hat{\Delta}_v^M(y_v^1, \dots, y_v^M) \tag{12}$$

*is a solution of the corresponding $M$-best-diverse problem* (4)

$$E^M(\{\boldsymbol{y}\}) = \sum_{i=1}^{M} E(\boldsymbol{y}^i) - \lambda \sum_{v \in \mathcal{V}} \Delta_v^M(y_v^1, \dots, y_v^M), \tag{13}$$

*where $\hat{\Delta}_v^M$ and $\Delta_v^M$ are related by* (11).

We will say that a vector $(y^1, \dots, y^M) \in (L_v)^M$ is *ordered*, if it holds $y^1 \leq y^2 \leq \dots \leq y^M$.

Given submodularity of $E$ the submodularity (an hence – solvability) of $E^M$ in (13) would trivially follow from the supermodularity of $\Delta^M$. However there hardly exist supermodular diversity measures. The ordering provided by Theorem 1 and the corresponding form of the ordering-enforcing diversity measure $\hat{\Delta}^M$ significantly weaken this condition, which is precisely stated by the following lemma. In the lemma we substitute $\infty$ of (11) with a sufficiently big values such as $C_\infty \geq \max_{\{y\}} E^M(\{y\})$ for the sake of numerical implementation. Moreover, this values will differ from each other to keep $\hat{\Delta}_v^M$ supermodular.

**Lemma 2.** *Let for any two **ordered** vectors $\boldsymbol{y} = (y^1, \ldots, y^M) \in (L_v)^M$ and $\boldsymbol{z} = (z^1, \ldots, z^M) \in (L_v)^M$ it holds*

$$\Delta_v(\boldsymbol{y} \vee \boldsymbol{z}) + \Delta_v(\boldsymbol{y} \wedge \boldsymbol{z}) \geq \Delta_v(\boldsymbol{y}) + \Delta_v(\boldsymbol{z}), \tag{14}$$

*where $\boldsymbol{y} \vee \boldsymbol{z}$ and $\boldsymbol{y} \wedge \boldsymbol{z}$ are element-wise maximum and minimum respectively. Then $\hat{\Delta}_v$, defined as*

$$\hat{\Delta}_v(y^1, \ldots, y^M) = \Delta_v(y^1, \ldots, y^M) - C_\infty \cdot \left[ \sum_{i=1}^{M-1} \sum_{j=i+1}^{M} 3^{\max(0, y^i - y^j)} - 1 \right] \tag{15}$$

*is supermodular.*

Note, eq. (11) and (15) are the same up to the infinity values in (11). Though condition (14) resembles the supermodularity condition, it has to be fulfilled for *ordered* vectors only. The following corollaries of Lemma 2 give two most important examples of the diversity measures fulfilling (14).

**Corollary 1.** *Let $|L_v| = 2$ for all $v \in \mathcal{V}$. Then the statement of Lemma 2 holds for* arbitrary $\Delta_v \colon (L_v)^M \to \mathbb{R}$.

**Corollary 2.** *Let $\Delta_v^M(y^1, \ldots, y^M) = \sum_{i=1}^{M-1} \sum_{j=i+1}^{M} \Delta_{ij}(y^i, y^j)$. Then the condition of Lemma 2 is equivalent to*

$$\Delta_{ij}(y^i, y^j) + \Delta_{ij}(y^i + 1, y^j + 1) \geq \Delta_{ij}(y^i + 1, y^j) + \Delta_{ij}(y^i, y^j + 1) \, for \, y^i < y^j \tag{16}$$

*and $1 \leq i < j \leq M$.*

*In particular, condition* (16) *is satisfied for the Hamming distance $\Delta_{ij}(y, y') = [\![y \neq y']\!]$.*

The following theorem trivially summarizes Lemmas 1 and 2:

**Theorem 2.** *Let energy $E$ and diversity measure $\Delta^M$ satisfy conditions of Lemmas 1 and 2. Then the ordering enforcing problem* (12) *delivers solution to the $M$-best-diverse problem* (13) *and is submodular. Moreover, submodularity of all non-unary potentials of the energy $E$ implies submodularity of all non-unary potentials of the ordering enforcing energy $\hat{E}^M$.*

## 4 Experimental evaluation

We have tested our algorithms in **two application scenarios**: (a) interactive foreground/background image segmentation, where annotation is available in the form of scribbles [3] and (b) Category level segmentation on PASCAL VOC 2012 data [9].

As **baselines** we use: (i) the sequential method `DivMBest` (2) proposed in [3, 25] and (ii) the clique-encoding `CE` method [19] for an (approximate) joint computation of $M$-best-diverse labelings. As mentioned in Section 2, this method addresses the energy $E^M$ defined in (4), however it has the disadvantage that its label space grows exponentially with $M$.

**Our method** that solves the problem (12) with the Hamming diversity measure (5) by transforming it into min-cut/max-flow problem [21, 28, 16] and running the solver [5] is denoted as `Joint-DivMBest`.

**Diversity measures** used in experiments are: the Hamming distance (5) `HD`, Label Cost `LC`, Label Transitions `LT` and Hamming Ball `HB`. The last three measures are higher order diversity potentials introduced in [25] and used only in connection with the `DivMBest` algorithm. If not stated otherwise, the Hamming distance (5) is used as a diversity measure. Both the clique encoding (`CE`) based approaches and the submodularity-based methods proposed in this work use only the Hamming distance as a diversity measure.

As [25] suggests, certain combinations of different diversity measures may lead to better results. To denote such combinations, the signs $\otimes$ and $\oplus$ were used in [25]. We refer to [25] for a detailed description of this notation and treat such combined methods as a black box for our comparison.

|  | M=2 | | M=6 | | M=10 | |
|---|---|---|---|---|---|---|
|  | quality | time | quality | time | quality | time |
| DivMBest | 93.16 | **0.45** | 95.02 | **2.4** | 95.16 | **4.4** |
| CE | **95.13** | 2.9 | **96.01** | 47.6 | **96.19** | 1247 |
| Joint-DivMBest | **95.13** | 0.77 | **96.01** | 5.2 | **96.19** | 20.4 |

Table 1: Interactive segmentation: per-pixel accuracies (quality) for the best segmentation out of $M$ ones and run-time. Compare to the average quality 91.57 of a single labeling. Hamming distance is used as a diversity measure. The run-time is in milliseconds (ms). `Joint-DivMBest` quantitatively outperforms `DivMBest`, and is equal to `CE`, however, it is considerably faster than `CE`.

## 4.1 Interactive segmentation

Instead of returning a single segmentation corresponding to a MAP-solution, diversity methods provide to the user a small number of possible low-energy results based on the scribbles. Following [3] we model only the first iteration of such an interactive procedure, i.e. we consider user scribbles to be given and compare the sets of segmentations returned by the compared diversity methods.

Authors of [3] kindly provided us their 50 graphical model instances, corresponding to the MAP-inference problem (1). They are based on a subset of the PASCAL VOC 2010 [9] segmentation challenge with manually added scribbles. Pairwise potentials constitute contrast sensitive Potts terms [4], which are submodular. This implies that (i) the MAP-inference is solvable by min-cut/max-flow algorithms [21] and (ii) Theorem 2 is applicable and the $M$-best-diverse solutions can be found by reducing the ordering preserving problem (12) to min-cut/max-flow and applying the corresponding algorithm.

**Quantitative comparison and run-time** of the considered methods is provided in Table 1, where each method was used with the parameter $\lambda$ (see (2), (4)), optimally tuned via cross-validation. Following [3], as a quality measure we used the per pixel accuracy of the best solution for each sample averaged over all test images. Methods `CE` and `Joint-DivMBest` gave the same quality, which confirms the observation made in [19], that `CE` returns an exact MAP solution for each sample in this dataset. Combined methods with more sophisticated diversity measures return results that are either inferior to `DivMBest` or only negligibly improved once, hence we omitted them. The run-time provided is also averaged over all samples. The max-flow algorithm was used for `DivMBest` and `Joint-DivMBest` and $\alpha$-expansion for `CE`.

**Summary.** It can be seen that the `Joint-DivMBest` qualitatively outperforms `DivMBest` and is equal to `CE`. However, it is considerably faster than the latter (the difference grows exponentially with $M$) and the runtime is of the same order of magnitude as the one of `DivMBest`.

## 4.2 Category level segmentation

The category level segmentation from PASCAL VOC 2012 challenge [9] contains 1449 validation images with known ground truth, which we used for evaluation of diversity methods. Corresponding pairwise models with contrast sensitive Potts terms of the form $\theta_{uv}(y, y') = w_{uv}[\![y \neq y']\!]$, $uv \in \mathcal{F}$, were used in [25] and kindly provided to us by the authors. Contrary to interactive segmentation, the label sets contain 21 elements and hence the respective MAP-inference problem (1) is not submodular anymore. However it still can be approximatively solved by $\alpha$-expansion or $\alpha$-$\beta$-swap.

Since the MAP-inference problem (1) is not submodular in this experiment, Theorem 2 is not applicable. We used two ways to overcome it. *First*, we modified the diversity potentials according to (15), as if Theorem 2 were to be correct. This basically means we were explicitly looking for ordered $M$ best diverse labelings. The resulting inference problem was addressed with $\alpha$-$\beta$-swap (since neither max-flow nor the $\alpha$-expansion algorithms are applicable). We refer to this method as to `Joint-DivMBest-ordered`. *The second* way to overcome the non-submodularity problem, is based on learning. Using structured SVM technique we trained pairwise potentials with additional constraints enforcing their submodularity, as it is done in e.g. [11]. We kept the contrast terms $w_{uv}$ and learned only a single submodular function $\hat{\theta}(y, y')$, which we used in place of $[\![y \neq y']\!]$. After the learning, all our potentials had the form $\theta_{uv}(y, y') = w_{uv}\hat{\theta}(y, y')$, $uv \in \mathcal{F}$. We refer to

| | MAP inference | M=5 | | M=15 | | M=16 | |
|---|---|---|---|---|---|---|---|
| | | quality | time | quality | time | quality | time |
| DivMBest | $\alpha$-exp[4] | 51.21 | **0.01** | 52.90 | **0.03** | 53.07 | **0.03** |
| HB* | HB-HOP-MAP[30] | 51.71 | - | 55.32 | - | - | - |
| DivMBest*$\oplus$HB* | HB-HOP-MAP[30] | - | - | 55.89 | - | - | - |
| HB*$\otimes$LC*$\otimes$LT* | LT$-$coop. cuts[17] | - | - | 56.97 | - | - | - |
| DivMBest*$\otimes$HB*$\otimes$LC*$\otimes$LT* | LT$-$coop. cuts[17] | - | - | - | - | 57.39 | - |
| CE | $\alpha$-exp[4] | **54.22** | 733 | - | - | - | - |
| CE$_3$ | $\alpha$-exp[4] | 54.14 | 2.28 | **57.76** | 5.87 | **58.36** | 7.24 |
| Joint-DivMBest-ordered | $\alpha$-$\beta$-swap[4] | 53.81 | 0.01 | 56.08 | 0.08 | 56.31 | 0.08 |
| Joint-DivMBest-learned | max-flow[5] | 53.85 | 0.38 | 56.14 | 35.47 | 56.33 | 38.67 |
| Joint-DivMBest-learned | $\alpha$-exp[4] | 53.84 | 0.01 | 56.08 | 0.08 | 56.31 | 0.08 |

Table 2: PASCAL VOC 2012. Intersection over union quality measure/running time. The best segmentation out of $M$ is considered. Compare to the average quality $43.51$ of a single labeling. Time is in seconds (s). Notation '-' correspond to absence of result due to computational reasons or inapplicability of the method. (*)- methods were not run by us and the results were taken from [25] directly. The MAP-inference column references the slowest inference technique out of those used by the method.

this method as to Joint-DivMBest-learned. For the model we use max-flow[5] as an exact inference method and $\alpha$-expansion[4] as a fast approximate inference method.

**Quantitative comparison and run-time** of the considered methods is provided in Table 2, where each method was used with the parameter $\lambda$ (see (2), (4)) optimally tuned via cross-validation on the validation set in PASCAL VOC 2012. Following [3], we used the Intersection over union quality measure, averaged over all images. Among combined methods with higher order diversity measures we selected only those providing the best results. The method CE$_3$ [19] is a hybrid of DivMBest and CE delivering a reasonable trade-off between running time and accuracy of inference for the model $E^M$ (4). Quantitative results delivered by Joint-DivMBest-ordered and Joint-DivMBest-learned are very similar (though the latter is negligibly better), significantly outperform those of DivMBest and only slightly inferior to those of CE$_3$. However the run-time for Joint-DivMBest-ordered and $\alpha$-expansion version of Joint-DivMBest-learned are comparable to those of DivMBest and outperform all other competitors due to use of the fast inference algorithms and linearly growing label space, contrary to the label space of CE$_3$, which grows as $(L_v)^3$. Though we do not know exact run-time for the combined methods (where $\oplus$ and $\otimes$ are used) we expect them to be significantly higher then those for DivMBest and Joint-DivMBest-ordered because of the intrinsically slow MAP-inference techniques used. However contrary to the latter one the inference in Joint-DivMBest-learned can be exact due to submodularity of the underlying energy.

## 5  Conclusions

We have shown that submodularity of the MAP-inference problem implies a fully ordered set of $M$ best diverse solutions given a node-wise permutation invariant diversity measure. Enforcing such ordering leads to a submodular formulation of the joint $M$-best-diverse problem and implies its efficient solvability. Moreover, we have shown that even in non-submodular cases, when the MAP-inference is (approximately) solvable with efficient graph-cut based methods, enforcing this ordering leads to the $M$-best-diverse problem, which is (approximately) solvable with graph-cut based methods as well. In our test cases (and there are likely others), such an approximative technique lead to notably better results then those provided by the established sequential DivMBest technique [3], whereas its run-time remains quite comparable to the run-time of DivMBest and is much smaller than the run-time of other competitors.

## Footnotes

[1]Pairwise binary potentials satisfying $\theta_f(0, 1) + \theta_f(1, 0) \geq \theta_f(0, 0) + \theta_f(1, 1)$ build an important special case of this definition.

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
