[Supplementary Material]

# Supplementary Materials: M-Best-Diverse Labelings for Submodular Energies and Beyond

*Proof of* **Theorem 1.** Let us consider the operation $\texttt{order}(\{\boldsymbol{y}\}, i, j)$, which takes a set of labelings $\{\boldsymbol{y}\} \in (L_{\mathcal{V}})^M$, two indices $i < j \in 1, \ldots, M$ and replaces labelings $\boldsymbol{y}^i$ and $\boldsymbol{y}^j$ by their node-wise minimum $\boldsymbol{y}^i \wedge \boldsymbol{y}^j$ and maximum $\boldsymbol{y}^i \vee \boldsymbol{y}^j$ respectively. As a result, this operation returns the new set of labelings:

$$(\boldsymbol{y}^1, \ldots, \boldsymbol{y}^{i-1}, \boldsymbol{y}^i \wedge \boldsymbol{y}^j, \boldsymbol{y}^{i+1}, \ldots, \boldsymbol{y}^{j-1}, \boldsymbol{y}^i \vee \boldsymbol{y}^j, \boldsymbol{y}^{j+1}, \ldots, \boldsymbol{y}^M). \tag{17}$$

In what follows we will show that

$$E^M \left( \texttt{order}(\{\boldsymbol{y}\}, i, j) \right) \leq E^M(\{\boldsymbol{y}\}). \tag{18}$$

Let $\{\boldsymbol{y}'\} = \texttt{order}(\{\boldsymbol{y}\}, i, j)$. Then $\{\boldsymbol{y}'\}_v$ is equal either to $(y_v^1, \ldots, y_v^i, \ldots, y_v^j, \ldots, y_v^M)$ or to $(y_v^1, \ldots, y_v^j, \ldots, y_v^i, \ldots, y_v^M)$. Since each $\Delta_v$ is permutation invariant, $\Delta^M(\{\hat{\boldsymbol{y}}'\}) = \Delta^M(\{\hat{\boldsymbol{y}}\})$. Summing it up with the following inequality, which follows from the submodularity of $E$,

$$\sum_{k=1}^{M} E(\boldsymbol{y'}^k) = \sum_{\substack{k=1 \\ k \neq i, k \neq j}}^{M} E(\boldsymbol{y}^k) + E(\boldsymbol{y}^i \wedge \boldsymbol{y}^j) + E(\boldsymbol{y}^i \vee \boldsymbol{y}^j) \leq \sum_{k=1}^{M} E(\boldsymbol{y}^k). \tag{19}$$

one obtains (18).

Assume the set of labelings $\{\hat{\boldsymbol{y}}\} = (\hat{\boldsymbol{y}}^1, \ldots, \hat{\boldsymbol{y}}^M)$ is a solution to (4):

$$\{\hat{\boldsymbol{y}}\} = \arg\min_{\{\boldsymbol{y}\}} E^M(\{\boldsymbol{y}\}). \tag{20}$$

Let us iteratively apply the operation $\{\hat{\boldsymbol{y}}\} := \texttt{order}(\{\hat{\boldsymbol{y}}\}, i, j)$ such, that indexes $i$ and $j$ follow the bubble-sort algorithm [1]. Each operation performs sorting for a single pair $i < j$ of indexes and due to (18) the energy $E^M\{\hat{\boldsymbol{y}}\}$ does not increase after the operation. As a result of the algorithm we obtain the ordered labeling set $\{\hat{\boldsymbol{y}}\}$ satisfying

$$E^M(\{\hat{\boldsymbol{y}}\}) \leq \min_{\{\boldsymbol{y}\}} E^M(\{\boldsymbol{y}\}), \tag{21}$$

which finalizes our proof.

$\square$

*Proof of* **Lemma 1.** Since $E$ is submodular and each $\Delta_v^M$ is permutation invariant we can apply Theorem 1 for $E^M$. This implies that $E^M$ has an ordered minimizer $\{\boldsymbol{y}^*\}$ and $\hat{E}^M(\{\boldsymbol{y}^*\}) = E^M(\{\boldsymbol{y}^*\})$.

Since the diversity controlling parameter $\lambda > 0$, the value of $-\lambda \hat{\Delta}_v^M(y^1, \ldots, y^M)$ is equal to $+\infty$ for an unordered set $(\boldsymbol{y}^1, \ldots, \boldsymbol{y}^M)$. Therefore, $\hat{E}^M(\{\boldsymbol{y}\})$ can be represented as follows:

$$\hat{E}^M(\{\boldsymbol{y}\}) = \begin{cases} E^M(\{\boldsymbol{y}\}), & \boldsymbol{y}^1 \leq \boldsymbol{y}^2 \leq \cdots \leq \boldsymbol{y}^M \\ \infty, & \text{otherwise} \end{cases}. \tag{22}$$

This implies $\arg\min_{\{y\}} \hat{E}^M(\{\boldsymbol{y}\}) \subseteq \arg\min_{\{y\}} E^M(\{\boldsymbol{y}\})$, which finalizes the proof. $\square$

*Proof of* **Lemma 2.** Let us consider $f(\boldsymbol{y}) = -\sum_{i=1}^{M}\sum_{j=i+1}^{M}\left(3^{\max(0,y^i-y^j)}-1\right)$. This potential is a sum of pairwise potentials $f_{ij}(y^i,y^j) = -\left(3^{\max(0,y^i-y^j)}-1\right)$. They are supermodular, which can be checked directly by definition. Moreover, by construction

$$f(\boldsymbol{y}\vee\boldsymbol{z}) + f(\boldsymbol{y}\wedge\boldsymbol{z}) = f(\boldsymbol{y}) + f(\boldsymbol{z}) \tag{23}$$

if either (i) both $\boldsymbol{y}$ and $\boldsymbol{z}$ are ordered vectors or (ii) $\boldsymbol{y}$ and $\boldsymbol{z}$ are comparable, i.e. $(\boldsymbol{y}\vee\boldsymbol{z},\boldsymbol{y}\wedge\boldsymbol{z})$ is either equal to $(\boldsymbol{y},\boldsymbol{z})$ or to $(\boldsymbol{y},\boldsymbol{z})$. Let us verify supermodularity of (15) by definition, i.e. for any $\boldsymbol{y}\in(L_v)^M$ and $\boldsymbol{z}\in(L_v)^M$, the following inequality has to be satisfied:

$$\hat{\Delta}_v(\boldsymbol{y}\vee\boldsymbol{z}) + \hat{\Delta}_v(\boldsymbol{y}\wedge\boldsymbol{z}) \geq \hat{\Delta}_v(\boldsymbol{y}) + \hat{\Delta}_v(\boldsymbol{z}). \tag{24}$$

For any ordered $\boldsymbol{y}\in(L_v)^M$ it holds $f(\boldsymbol{y}) = 0$. Therefore, taking into account (14), the inequality (24) holds for any ordered $\boldsymbol{y}$ and $\boldsymbol{z}$. For any comparable $\boldsymbol{y}$ and $\boldsymbol{z}$ the inequality (24) is trivial. For any other $\boldsymbol{y}$ and $\boldsymbol{z}$ the following strict inequality holds $f(\boldsymbol{y}\vee\boldsymbol{z}) + f(\boldsymbol{y}\wedge\boldsymbol{z}) > f(\boldsymbol{y}) + f(\boldsymbol{z})$. This implies that for a sufficiently big $C_\infty$, the inequality (24) holds for arbitrary $\Delta_v(y^1,\ldots,y^M)$. $\square$

*Proof of* **Theorem 2.** Since energy $E$ and diversity measure $\Delta^M$ satisfy conditions of Lemma 1, the ordering enforcing problem (12) delivers solution to the $M$-best-diverse problem (13). Moreover, since each component $\Delta_v^M$ of $\Delta^M$ satisfies conditions of Lemma 2, the function $\hat{\Delta}^M$ is supermodular and $-\hat{\Delta}^M$ is submodular. Since energy $E$ is submodular either, the ordering enforcing energy $\hat{E}^M$ is submodular as sum of submodular functions. $\square$

## References

[1] T.H. Cormen, C.E. Leiserson, R.L. Rivest, C. Stein. *Introduction to algorithms third edition.* 2009.