[Reviews · NeurIPS 2015]

Submitted by Assigned_Reviewer_1

I really like this paper, as it shows how we can use existing algorithms to jointly optimize over several solutions, which will be both of low-energy and diverse. I think Lemmas 1 and 2 themselves are very important observations and the resulting trick of "submodularizing" the original function (while keeping the same optima) is very clever.

I think that the discussion around Lemma 2 can be significantly improved if the authors provide an explicit example (with exact numbers) of a small model, and how it gets converted afterwards (even for the binary case with M=3). A description of the new graph on which you run graph cut would be very illustrative.

I did not check the proofs in much detail. Could the authors give a brief proof why the strict inequality on L557-558 holds?
Summary: In this work the authors show how one can choose a set of K configurations that have a high energy *and* are diverse using existing, well-established, MAP computation techniques. I strongly recommend this paper for acceptance.

Submitted by Assigned_Reviewer_2

The paper considers the problem of finding best diverse solutions of energy minimisation problems with focus on submodular energy functions. The authors further present an alternative formulation of the problem as an ordering enforcing problem. Furthermore, the proposed methods are compared to other state of the art methods in experiments.

Quality. The presented results are interesting and seem to be correct. However, I'm not sure of the implications of Lemma 2. By replacing the original diversity measure by the one in (15), supermodularity of the diversity measure is achieved. What effects does this replacement have? Do the characteristics of the diversity measure change? Furthermore, regarding Corollary 2, it would be interesting which other diversity measures satisfy the given condition. Regarding the experiments, the authors seem to report the best result from all M obtained labelings. Wouldn't it make sense to also report the average quality and the corresponding standard deviation?

Clarity. The clarity of the manuscript is good, but could be improved at several points: At several points, e.g. in the appendix, the authors refer to "the algorithms", however it is unclear to me what exactly they refer to. In equations (11) and (15) the authors use the same notation for different quantities. It would be interesting to explicitly state how these two diversity measures are related and what are the implications for the obtained diverse labelings when using the one or the other.

Originality. The paper improves results for identifying diverse solutions jointly over a recently submitted paper, especially by approaching runtime/complexity issues. While there is significant overlap, this improvements seem sufficiently interesting and original to me.

Significance. The presented results could be interesting for many applications in many different fields.

Further comments: * L152: "This implies that in this case" * L184: "The corresponding factor ..." * L196: "compared to" * L 237: "If E is submodular, then" * L 254: "This order can be enforced" * Table 1: State meaning of "M" again. * Provide details on how the cross-validation for selecting lambda is performed. * L404: "the intersection over union quality measure" ... what does this mean? * L430: "much smaller than" * Is there any reasonable approach to learn which diversity measures to combine? * Please carefully check the grammar of the supplement. * several places: "there holds" => "it holds"
Summary: The submitted paper presents interesting ideas for identifying diverse labelings but lacks some clarifications and details.

Submitted by Assigned_Reviewer_3

This paper studies the M-best-diverse problem, an extension of MAP problem, which seeks M > 1 diverse configurations minimising a given energy function. The paper proves that if the original energy function is submodular and the diverse measure is nodewise, then the M-best-diverse problem can be converted to an ordering enforcing M-best-diverse problem (Theorem 1), which is submodular under some conditions and thus can be minimised efficiently with min-cut / max-flow

(Theorem 2).

Pros * The paper is generally well written but there are some points which may need further clarification (detailed below).

* The M-best-diverse problem it addresses is both interesting and useful theoretically and practically.

* The approach the authors take is novel and interesting and seems technically sound.

Cons * Some points may need clarification:

-- Before lemma 2, ^ and v operators are defined for two labelling but in (14) they are applied to two labelling sets (i.e. two sets of M labellings). The overloading of these operators may need to be explicitly explained.

-- Line 169-170, factors in the new graph are a union of a new factor and original factors. The new factor should be a function on all the new nodes not the new nodes themselves.

-- Line 176-177, please be more precise on the definition of "easy".

-- Line 186-187, similar to previous comment, "bad" is vague here.

-- Line 313-314, please make it clear what method is used even if it may be an off-the-shelf method.

-- Experiment: it would be good to mention sizes of the problems.

Summary: The paper proposes a novel algorithm on the M-best-diverse problem for graphical models. It is likely to have good impact.

Submitted by Assigned_Reviewer_4

The authors propose an efficient method for jointly inferring M-Best Diverse solutions for MRFs with submodular energies, along with theoretical justification. While the manuscript is well-written and the presentation is clear, novelty is reduced by the ICCV submission. There is significant overlap between these manuscripts, in particular the introduction and related work sections read nearly verbatim and the sole figure of this manuscript is identical to one in the ICCV paper.

The experimental setup is confusing since the main point is one about diversity, but reporting average quality masks this.

Similar high-quality solutions would produce a high average quality score, although their diversity is low.

It is also unclear how average quality is computed, the authors should clarify this; I assume the average is over samples and the M results?

Finally, no significance claims can be made since confidence intervals are lacking on all results.

A minor comment is that proofs or proof sketches of the main results (Thm 1 & 2) should be incorporated into the main text. The caption for Table 2 can be reduced significantly to make room for these by not reiterating identical points in the main text.

Some minor detailed comments: * Define notation as it is used, rather than in one block beginning in Sec. 2. * Line 222 states that for binary MRFs submodularity of energy and potentials is bidirectional. Provided a reference. * In the proof of Thm 1 it is not clear what is meant by "incompatible". * In references, capitalize acronyms (e.g. "SVM" on Line 455) and keep author initials consistent (e.g. Y. Boykov vs. Y. Y. Boykov). * Missing $_v$ subscript in Eq. (6)
Summary: The manuscript is well-written, the approach well-motivated and strong theoretical justification is provided. Novelty is diminished by a parallel ICCV submission with significant overlap. Experimental results are adequate but the chosen metric masks diversity in the solution set for each approach, which is the ultimate point of the paper.

Author Feedback
Author rebuttal: We thank all reviewers for their positive and constructive comments. We address the main concerns below. Small edits will be incorporated into the camera-ready version.

R1: "In eq. (11) and (15) the authors use the same notation for different quantities"

Note that (11) and (15) are indeed the same up to the infinity values in (11), which are substituted with the very large numbers to keep the diversity measure supermodular. This allows for a more straightforward implementation, since infinities typically can not be encoded in a program. See also comment in L276.

R1: "I'm not sure of the implications of Lemma 2. By replacing the original diversity measure by the one in (15), supermodularity of the diversity measure is achieved. What effects does this replacement have? Do the characteristics of the diversity measure change?"

Replacing the original diversity measure by the one in (15) results in the *same* optimal M labellings, as with the original diversity. This is stated by Lemma 1 up to the relation between (11) and (15) explained above. We will comment on this in the paper.

R1: "regarding Corollary 2, it would be interesting which other diversity measures satisfy the given condition"

In particular, this holds for convex functions of y^j-y^i, e.g. min(|y^j-y^i|, a) and |y^j-y^i|^p for p<=1. We will comment on this in the paper.

R1: "Wouldn't it make sense to also report the avg. quality...?"

In our work we follow the experimental setup from [4, 26] to keep our results comparable to those ones.
Indeed, we are not sure whether the averaging over M diverse solutions would make sense. We will check this point.

R1: "At several points, e.g. in the appendix, the authors refer to "the algorithms", however it is unclear to me what exactly they refer to"

Thanks, we will add the corresponding references. In particular, the algorithm mentioned in appendix is in L520-526.

R2: "The paper introduces a sub modular approach for diversity of M-best solutions. Its contribution, beyond its supplementary material, is incremental"

With all respect, our algorithm is qualitatively close or even equal to the best known approach [2] and is several orders of magnitude faster. In the same time it noticeably outperforms the original approach [4] and is only few times slower.

R3: "There is significant overlap between these manuscripts, in particular the introduction and related work sections read nearly verbatim and the sole figure of this manuscript is identical to one in the ICCV paper"

We agree, that is because both papers address nearly the same problem and the related work did not change since the first submission. The sole figure is duplicated to keep our paper self-contained. Note, that nevertheless the contribution of this paper significantly differs from the one in the ICCV submission: the ICCV paper formulates a problem to be solved, whereas the current work proposes an efficient solution to the formulated problem.

R3: "The exp. setup is confusing since the main point is one about diversity, but reporting avg. quality masks this. Similar high-quality solutions would produce a high avg. quality score, although their diversity is low. It is also unclear how avg. quality is computed, the authors should clarify this; I assume the average is over samples and the M results?"

Thanks for pointing the lack of description in the paper. We will fix this. In our work we follow the experimental setup from [4, 26] to keep our results comparable to those ones. For interactive segmentation experiment we took the best per pixel accuracy among M solutions for each sample and then averaged them over the samples. For category level segmentation we again took the best solutions for each sample according to IoU and then calculated the IoU over all dataset.

R4: Thanks for the number of spotted inaccuracies. We will fix all of them.

R5: "I think that the discussion around Lemma 2 can be significantly improved..."
Thanks, we will do our best to improve it.

R5: "Could the authors give a brief proof why the strict inequality on L557-558 holds?"
The strict inequality in L557-558 follows from the inequality:
f_{i,j}(y^i, y^j) + f_{i,j}(y^i+1, y^j+1) > f_{i,j}(y^i+1, y^j) + f_{i,j}(y^i, y^j+1) for y^i >= y^j,
which holds since f_{i,j}(y^i, y^j) is strictly concave for y^i >= y^j as a function of y^i - y^j. We refer to [29, eq.(5) and (6)] for more details for the moment. We will add the corresponding comments to our paper.

R6: "The novelty of this paper is mainly in the proof of the supermodularity of the diversity term, which is rather simple"

On our view, the novelty of the paper is in the fact that the M-best-diverse problem can be equivalently reformulated as a submodular one, given the MAP-inference problem is submodular and certain conditions on the diversity measure hold. The proof of supermodularity, which is mentioned, only defines these certain conditions.